# Comparative Analysis of Production Possibility Frontier in Measuring Social Efficiency with Data Envelopment Analysis: An Application to Airports

**Tae-Woong Yang [1], Joon-Ho Na [2] and Hun-Koo Ha [1],*** 

[1]  Graduate School of Logistics, Inha University, Incheon 22212, Korea; taewyang@gmail.com
[2]  Antai College of Economics and Management, Shanghai Jiao Tong University, Shanghai 200030, China; njh5020@sjtu.edu.cn
*  Correspondence: hkha@inha.ac.kr; Tel.: +82-32-860-8232

**Abstract:** The environmental sustainability is a globally important issue, particularly in the global warming. There are many institutions and people who are interested in the greenhouse gases emissions issue and policies that attempt to improve the problem. The aviation industry is not an exception. Under this background, there has been much research on airport efficiency analysis, undesirable outputs, and on evaluating productivity with respect to environmental factors. In the efficiency analysis models with the undesirable outputs in the airports, there are two main types of production possibility frontiers. The first type is the frontier based on the Shephard technology, which involves a weak-disposability concept, using a single abatement factor. The second one is the frontier on the Lozano-Gutiérrez technology, which tries to take the weak disposability into account by regarding the undesirable outputs as input. However, they have limitations. Additionally, no study has provided how to apply weak disposability correctly. To find out the limitations and give standard to utilize weak disposability, we compare models with two issues that must be scrutinized. In this paper, we show that these two concepts have limitations in making the production possibility area. To overcome this limitation, we propose an undesirable-output model using multiple abatement factors based on weak disposability with the slack-based measure (SBM) approach. We analyze, comparatively, the different social efficiency performances according to two issues among the three approaches in estimating production possibility frontier, using the Shepard model, the Lozano-Gutierrez model, and our proposed model. To provide correct standard of measurement and apply characteristics of undesirable outputs, we study not only theoretically, but also empirically, with data from Korea's 13 domestic airports.

**Keywords:** ecoefficiency; undesirable output; efficiency analysis; slack-based measure; airport ecoefficiency

## 1. Introduction

Environmental issues are of crucial importance, and therefore, many institutes and countries have implemented strict regulation and management criteria. Above air pollution and greenhouse gas (GHG) emissions are the most actively discussed issues. While GHGs primarily include carbon dioxide ($CO_2$), methane ($CH_4$), nitrous oxide ($N_2O$), water vapor, etc., reduction in $CO_2$ emissions is the main objective of policies, because $CO_2$ constitutes about 77% of total GHGs emissions [1]. Transport section takes charge of 13% of total GHGs emissions. In that, Aviation takes 13% of transport's $CO_2$ emissions [2]. Independently, air transportation accounts for 2% of total $CO_2$ emissions [3]. Moreover, because of being adjacent to atmosphere, there is a viewpoint that air transportation influences on air pollution more critically than other fields. These stand for the necessity of managing $CO_2$ emissions in

air transportation. In this situation, eco-efficiency, which has the ability to evaluate the efficiency of production considering environmental factors, plays a role. Previous studies have suggested various standards for estimating eco-efficiency based on models using undesirable outputs. Such models have been used to estimate efficiency by taking into account the harmful side effects of outputs in aviation's case, complaints, flight delays, noise, pollution, etc.

In models using the data envelopment analysis (DEA) method, including undesirable-output models, a production possibility function (PPF) consists of constraint in evaluating efficiency. To measure efficiency correctly, the correct PPF is needed, because estimated boundary by a PPF determines degree of efficiency. It can be said that correct evaluation of efficiency is influenced by how accurately a PPF is estimated. Undesirable-output models depend on perspectives concerning undesirable outputs and how to construct PPFs using the perspectives. Previous studies measuring efficiency in the aviation industry have been limited in terms of estimating the production possibility area.

Previous studies in aviation industry have estimated the production possibility area of undesirable-output models based mainly on two perspectives. First one is the Shephard technology, which consider weak disposability as a characteristic of undesirable output. Weak disposability is a concept of relation between desirable output and undesirable output, that is, these two outputs produce or decrease together. However, the Shephard technology has limitations in terms of practicality and a violation of convexity axiom by using a single abatement factor. Secondly, Lozano and Gutiérrez suggest a model (Lozano-Gutiérrez model) that is based on the perspective concerning undesirable outputs as input, which takes weak disposability into account [4]. However, the hybrid model fails to reflect weak disposability. In spite of the limitations, previous studies accept the perspectives without question.

In this context, this paper compares perspectives and undesirable-output models mainly used in the aviation industry and suggests a model that mitigates limitations of conventional models by employing weak disposability and multiple abatement factors (Abatement factor is a variable which allows outputs to reduce to a decreased level of production activity and it forces desirable and undesirable outputs to contract together, when the level of a production activity decrease). The results based on Korean airports indicate that the proposed model with the slack-based measure (SBM) approach evaluates eco-efficiency better than conventional models. The SBM based model has an advantage in that the approach has no directionality to find a benchmark point and it makes the model includes characteristics of undesirable outputs.

The paper is organized as follows: Section 2 provides a review of previous research, not only on aviation, but also on other related technologies. Section 3 employs PPFs and areas to compare undesirable-output models in the context of the aviation industry. There are two issues that should be considered regarding using models. According to the issues, the comparison will be provided. We consider the precondition for an undesirable-output model and propose a modified model as a standard of correct constraint. To represent practical analysis of former discussion, with the same issues as Section 3, Section 4 presents the results for the case of Korean airports based on the proposed model and compares the results between models. Section 5 provides conclusion.

## 2. Literature Review

Many studies have considered undesirable-output models using the DEA method, including studies that consider characteristics of undesirable outputs and production possibility areas. An incorporated measurement of productivity with the environmental factor is done by Pittman (1983) [5]. Färe et al. (1989) claim that Shephard (1970) constructs the Shephard technology using weakly disposable undesirable outputs in a PPF. Fare et al. (1993), Chung et al. (1997), and Picazo-Tadeo and Prior (2005) apply weak disposability to directional distance function with undesirable outputs. Additionally, Zhou et al. (2006, 2007) employ weak disposability based on non-radial approach [6–12]. Haliu and Veeman (2001) regard undesirable outputs as input, asserting that weak disposability makes the essence of undesirable outputs ambiguous [13]. By contrast, Färe and Grosskopf (2003) argue that

undesirable outputs defined as input, as asserted in Haliu and Veeman, cannot reflect the nature of undesirable outputs [14]. While Seiford et al. (2002) provide other perspectives using a linear monotone decreasing transformation on the basis of BBC to reflect the undesirable outputs [15], it cannot reflect weak disposability between desirable and undesirable outputs.

Kuosmanen (2005) points out some limitations of the Shephard technology coming from a single abatement factor and provides the Kousmanen technology as an alternative with multiple abatement factors [16]. Färe and Grosskopf (2009) argue that the Kuosmanen technology is not a correct production possibility function because it overestimates the area [17]. Kuosmanen and Podinovski (2009) verify that the Shephard technology has an underestimation problem and that the Kuosmanen technology is the correct PPF that fully satisfies Shephard's definition and assumption of weak disposability [18].

In the aviation industry, there have been many attempts to consider undesirable outputs when efficiency, based on the DEA method, is evaluated. Yu (2004) evaluates the efficiency of Taiwanese airports by using the directional distance function (DDF) and the window analysis with the DEA method by factoring in aircraft noise as an undesirable output [19]. Yu uses the Shephard technology as a constraint to reflect weak disposability. Pathomsiri et al. (2008) use the DDF approach to consider delays as undesirable outputs and the Shephard technology as a constraint to assess the productivity of U.S. airports [20]. Lozano and Gutiérrez (2011) produce an undesirable-output model to assess the efficiency of Spanish airports and include delays as an undesirable output based on the SBM [4] (related studies on airlines include, for example, Ha et al. (2011a) and Scotti and Volta (2015) [21,22]) approach. Like Hailu and Veeman, they regard undesirable outputs as input, include the slack of undesirable outputs, and reflect weak disposability by setting a single abatement factor. To evaluate the efficiency of Korean airports, while considering $CO_2$ emissions as an undesirable output, Ha, H.K. (2011b) uses the perspective of Hailu and Veeman on the undesirable-output equation to set the slack of undesirable outputs [23]. Fan et al. (2014) measure the efficiency of Chinese airports by employing the Shephard technology as a constraint on the CRS assumption with flight delays [24] (related studies on airlines include, for example, Ha et al. (2011a) and Scotti and Volta (2015) [21,22]).

## 3. Methodology

### 3.1. Comparison of Perspectives on Undesirable Outputs (Weak Disposability Issue)

There are two issues in manifesting undesirable output on PPF. We address them in Sections 3.1 and 3.2, respectively. The first one is about how to definite characteristics of undesirable outputs, that is, weak disposability issue. It is not only the matter of equality or inequality in undesirable output restriction, but also the matter of reflecting weak disposability or not and how to reflect it. Following perspectives are included in this issue. According to the selection of a perspective, PPF is differently constructed.

#### 3.1.1. Undesirable Output as Input (Input Perspective)

Undesirable outputs generate from production activity such that they are associated with some desirable outputs. Undesirable outputs have a similar property to that of inputs, that is, the lower the level of undesirable outputs produces, the better the production is. Haliu and Veeman (2001) put undesirable outputs into a PPF as inputs [13]. The related axiom is as follows:

Axiom. If $(v, w, x) \in Y$ and $v \geq v'$, $w' \geq w$, $x' \geq x$, then $(v', w', x') \in Y$.

$Y$ is a technology that produces a desirable output $v$ and an undesirable output $w$ from input $x$. In this technology, undesirable outputs and inputs have the same inequality, which means they have the same sign of slack. In this case, undesirable outputs influence its efficiency score, particularly under the SBM approach(Slacks-based measure(SBM) is sub-method in DEA provided by Tone(2001) [25], and can be improved. Lozano and Gutiérrez (2011) and Ha, H.K. (2011b) employs this idea as follows [4,23]:

$$\sum_{k=1}^{K} z_k w_{kj} \leq w_j, \; j = 1, \ldots, J. \tag{1}$$

Equation (1) is an equation of an undesirable output w. The undesirable-output inequality is the same as the input inequality. This constraint implies that both inputs and undesirable outputs are unfavorable. Haliu and Veeman do not accept weak disposability for three reasons. First, the use of an equality restriction can produce a reference set and can thus substantially inflate the efficiency score. Second, weak disposability gives an undesirable output an undetermined effect on efficiency. Third, undesirable outputs, such as pollution, have negative shadow prices [13]. Färe and Grosskopf (2003) refute the criticism of Haliu and Veeman, but there are additional issues to consider, which we address later on. Färe and Grosskopf (2003) point out that the perspective of Haliu and Veeman is not realistic. If the inequality is satisfied then an infinite output, namely a bad output, can be generated from a finite input [14]. Then, the property of an undesirable output cannot be considered a by-product of a desirable output. Due to this clear limitation, the perspective barely employed in previous research. Therefore, in comparison, we will only employ the critical view of the perspective. Instead of this perspective, hybrid version of the input perspective and weak disposability perspective, namely, the hybrid perspective, will be discussed.

### 3.1.2. Undesirable Output as Fixed Value (Weak Disposability Perspective)

If an innovative technology that can reduce pollutants or other harmful products, such as purifying facilities or equipment, is created in an industry, then undesirable outputs can be reduced. This paper assumes that production activities are at the same level as the pollutant reducing technology. To reduce undesirable outputs, desirable outputs also have to be reduced. The Shephard technology for undesirable outputs includes the property that reduces the amount of outputs through weak disposability.

Shephard (1970) defines weak disposability of outputs as follows: if $(v, w) \in P(x)$ and $0 \leq \theta \leq 1$ then $(\theta v, \theta w) \in P(x)$. If $Y$ is weakly disposable, then, given input $x$, production outputs $(v, w)$ can be downsized together by a factor of theta.

The Shephard technology axioms are as follows:

> Axiom 1. If $(v, w, x) \in Y$, $0 \leq v' \leq v$ and $x' \geq x$, then $(v', w, x') \in Y$,
> Axiom 2. If $(v, w, x) \in Y$, $\theta \in [0; 1]$, then $(\theta v, \theta w, x) \in Y$,
> Axiom 3. $(v, w) \in Y$, $w = 0$ implies that $v = 0$,
> Axiom 4. Convexity.

Axiom 1 shows that an input x and a desirable output v are freely disposable. Axiom 2 shows that outputs $(v, w)$ are weakly disposable. If desirable and undesirable outputs have null-jointness, then axiom 3 is satisfied. If the production activity generates both undesirable and desirable outputs, and if there are no undesirable outputs, then there are no desirable outputs because the production activity has stopped. Axiom 4 is a basic axiom of PPFs in DEA method that can explain $Y$ as convex [7]. The formula of the Shephard technology employs the VRS assumption is as follows:

$$
P^S(x) = \left\{
\begin{array}{l}
(v, w): \\
\text{(a)} \; \theta \sum\limits_{k=1}^{K} z_k v_{km} \geq v_m, m = 1, \ldots, M, \\
\text{(b)} \; \theta \sum\limits_{k=1}^{K} z_k w_{kj} = w_j, j = 1, \ldots, J, \\
\text{(c)} \; \sum\limits_{k=1}^{K} z_k x_{kn} \leq x_n, n = 1, \ldots, N, \\
\text{(d)} \; z_k \geq 0, k = 1, \ldots, K, \\
\text{(e)} \; \sum\limits_{k=1}^{K} z_k = 1, \; 0 \leq \theta \leq 1
\end{array}
\right\}. \tag{2}
$$

*k*: number of airports

*m*: number of desirable outputs

*j*: number of undesirable outputs

*n*: number of inputs

$v_{km}$: observed amount of desirable output *m* of airport *k*

$w_{kj}$: observed amount of undesirable output *j* of airport *k*

$x_{kn}$: observed amount of input *n* of airport *k*

$v_m$: specific observation value of desirable output *m*

$w_j$: specific observation value of undesirable output *j*

$x_n$: specific observation value of input *n*

$\theta$: single abatement factor

In Equation (b) in (2), if the equality exists instead of an inequality, then this implies no disposability. An abatement factor and an equality is necessary to impose weak disposability. If the level of production activity changes, then outputs will also change, which is referred to as an abatement effort. Then, the abatement factor, appearing as $\theta$ in Equation (2), allows outputs to be reduced to a decreased level of production activity and forces desirable and undesirable outputs to contract together.

Figure 1 compares the production possibility area. If there are three observations A, B, and C, then the area bounded by $\overline{aA}$, $\overline{AB}$, and the horizontal line originating from point B is of the Haliu and Veeman technology. The area bounded by $\overline{0A}$, $\overline{AB}$, $\overline{BC}$, and $\overline{Cc}$ is of the Shephard technology ($P^S(x)$).

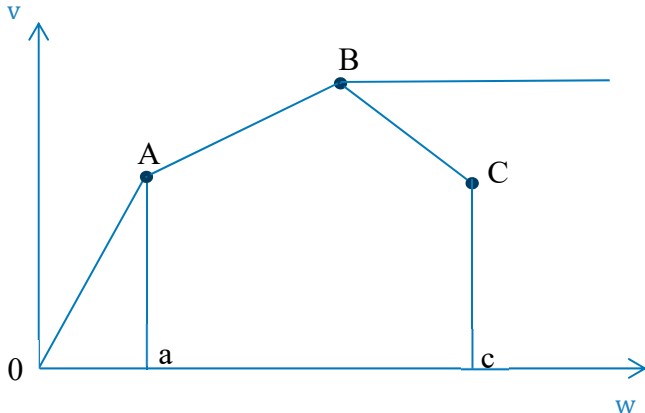

**Figure 1.** A comparison of production possibility areas.

As stated earlier, Haliu and Veeman point out the limitations of weak disposability [13]. Färe and Grosskopf (2003) refute Haliu and Veeman, but their argument is limited. In this regard, the following supplements and modifications are provided. First, "[A] weak disposability makes undesirable output[s] leave [an] undetermined effect on efficiency." Färe and Grosskopf state that this problem is related to how efficiency is measured and thus is not a matter of a reference technology but a problem with the entire model. A detailed explanation is provided later. Second, "undesirable output[s] like pollutant[s] have negative shadow price[s]." Färe and Grosskopf insist that because a direction can be imposed on variables in the DDF, the efficiency of effects on negative shadow prices and undesirable outputs can be taken into account [14]. In Figure 2, if there is an inefficient observation e, then the arrow $d_1$, which is the direction of projection of e, and $d_1$, have a contradiction. The direction of arrow $d_1$ is decided by the DDF model, which uses Shephard technology. A DDF that increases desirable outputs, while reducing undesirable outputs is problematic. Undesirable outputs with weak disposability cannot be decreased by themselves. Undesirable outputs can only be decreased if the decrease involves a decrease in desirable outputs, but the DDF allows undesirable outputs to decrease alone.

Figure 2 shows an inefficient point e. The projected direction of $d_1$ employed by the DDF allows the model to evaluate the efficiency of e. Unfortunately, $d_1$ implies that desirable outputs increase when

undesirable outputs decrease, which contradicts the assumption of weak disposability implied by the Shephard Technology. Undesirable outputs not only have negative shadow prices, as pointed by Haliu and Veeman, but also a productive property. Undesirable outputs represent a variable whose nature differs from that of inputs or desirable outputs. It is not accurate to define undesirable outputs in the same way as inputs, and therefore, undesirable outputs are recognized as a variable with properties of both inputs and desirable outputs. Weak disposability is not a means to make undesirable outputs undetermined (as Haliu and Veeman [13]), but it is a property that represents undesirable outputs as a different variable.

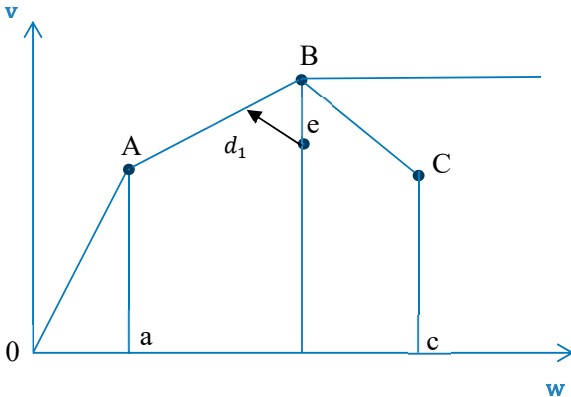

**Figure 2.** The direction of the DDF projection from Färe and Grosskopf (2003).

In the absence of a DDF, an increase in desirable outputs and a decrease in inputs must be selected in the production of the same amount of undesirable outputs (e.g., a movement from point e to the benchmark point B). This implies that the equality affects the efficiency score in terms of the whole measurement.

3.1.3. Undesirable Output as Hybrid Variable (Hybrid Perspective)

Lozano and Gutiérrez (2011) reflect weak disposability by using an abatement factor as well as null-jointness (Lozano and Gutiérrez (2011) and Ha, H.K. (2011b) represents similar PPFs [4,23]. This paper considers the model in Lozano and Gutiérrez representatively because this is the first to take these types of PPF and SBM approaches,), and introduce weak disposability perspective on the Haliu and Veeman technology. That is, they attempt to reflect a mixture of characteristics of inputs and outputs in undesirable outputs. Equation (3) is the PPF of the Lozano and Gutiérrez model, which includes the VRS assumption:

$$
P^L(x) = \left\{
\begin{array}{l}
(v,w): \quad \theta \sum_{k=1}^{K} z_k v_{km} \geq v_m, m = 1, \ldots, M, \\
\quad \theta \sum_{k=1}^{K} z_k w_{kj} \leq w_j, j = 1, \ldots, J, \\
\quad \sum_{k=1}^{K} z_k x_{kn} \leq x_n, n = 1, \ldots, N, \\
\quad z_k \geq 0, k = 1, \ldots, K, \\
\quad \sum_{k=1}^{K} z_k = 1, \ 0 \leq \theta \leq 1
\end{array}
\right\}.
\tag{3}
$$

$k$: number of airports

$m$: number of desirable outputs

$j$: number of undesirable outputs

$n$: number of inputs

$v_{km}$: observed amount of desirable output $m$ of airport $k$

$w_{kj}$: observed amount of undesirable output $j$ of airport $k$

$x_{kn}$: observed amount of input $n$ of airport $k$

$v_m$: specific observation value of desirable output $m$ desirable output $m$

$w_j$: specific observation value of undesirable output $j$

$x_n$: specific observation value of input $n$

$\theta$: single abatement factor

The abatement factor $\theta$ causes desirable and undesirable outputs to move together, but $\theta$ cannot satisfy weak disposability by itself. An abatement factor can make only the output move proportionally. In the Shephard technology, undesirable outputs have no directional nature because of an equality in (b) found in Equation (2). In this case, the abatement factor can let undesirable outputs follow desirable outputs if the level of production activity remains unchanged. In the case of the directional nature of undesirable outputs, as in the case of inputs, changes in desirable and undesirable outputs are proportional but in different directions. This implies that they never move together and do not have weak disposability. In this regard, Lozano and Gutiérrez combine the advantages of the Haliu and Veeman technology and the assumption of weak disposability, but as long as the inequality or slack of undesirable outputs (the equation with inequality can be transposed as that of equality by using slack.), which can move independently, exists, weak disposability cannot be reflected in the measurement. In terms of the whole measurement based on the SBM approach, a projection may head toward a benchmark point with only less undesirable outputs. Lozano and Gutiérrez fail to consider weakly disposable undesirable outputs. To impose weak disposability on the technology, the equality and the abatement factor are required. Figure 1 shows the production possibility area of the Lozano and Gutiérrez technology. The area bounded by $\overline{0A}$, $\overline{AB}$, and the horizontal line from B is the area under the Lozano and Gutiérrez technology. Based on a comparison of production possibility areas, the Lozano and Gutiérrez technology shows an area different from that of the Shephard technology.

*3.2. Comparison of Perspectives on the Abatement Factor (the Abatement Factor Issue)*

Second is the abatement factor issue. The abatement factor ($\theta$) has a crucial influence on realizing weak disposability between desirable and undesirable outputs by making them share an impact through it. In this paragraph, we discuss about problems from misused abatement factor in conventional models and how to correctly reflect it.

Yu (2004), Pathomsiri et al. (2008), and Fan et al. (2014) employ the Shephard technology as constraints in their models [19,20,24]. The Shephard technology imposes a single abatement factor on all observations. It has two key limitations. In Equation (2), θ is a single abatement factor that makes all observations or production activities have the same abatement effort. This means that if there are three observations, production activities, or three decision-making units (DMUs), for example, firms A, B, and C, then these firms reflect the same proportion when they control their own level of production activity. Three abatement factors $\theta^A$, $\theta^B$ and $\theta^C$ should exist. It is not practical to set the same abatement factor for all firms. In addition, the Shephard technology has a limitation in convexity. Kuosmanen and Podinovski (2009) and Podinovski and Kuosmanen (2011) verify the violation of the axiom of the Shephard technology [18,26].

$$P^K(x) = \left\{ \begin{array}{l} (v,w) : \sum\limits_{k=1}^{K} \theta^k z_k v_{km} \geq v_m, m = 1, \ldots, M, \\ \sum\limits_{k=1}^{K} \theta^k z_k w_{kj} = w_j, j = 1, \ldots, J, \\ \sum\limits_{k=1}^{K} z_k x_{kn} \leq x_n, n = 1, \ldots, N, \\ z_k \geq 0, k = 1, \ldots, K, \\ \sum\limits_{k=1}^{K} z_k = 1, \ 0 \leq \theta \leq 1 \end{array} \right\}. \tag{4}$$

$k$: number of airports

$m$: number of desirable outputs

$j$: number of undesirable outputs

$n$: number of inputs

$v_{km}$: observed amount of desirable output $m$ of airport $k$

$w_{kj}$: observed amount of undesirable output $j$ of airport $k$

$x_{kn}$: observed amount of input $n$ of airport $k$

$v_m$: specific observation value of desirable output $m$

$w_j$: specific observation value of undesirable output $j$

$x_n$: specific observation value of input $n$

$\theta^k$: abatement factors

Kuosmanen (2005) provides the Kuosmanen technology by using multiple abatement factors to address the limitations. With multiple abatement factors, each observation can have an abatement effort in the Kuosmanen technology [16]. Based on the CRS assumption, $\theta$ and $\theta^k$ are equal to one (As suggested in Färe and Grosskopf (2003) and Shephard (1974) [14,27]), if a model has weak disposability by setting the same abatement factor, then desirable (good) and undesirable (bad) outputs can contract together in a PPF. This implies that the PPF has an equality sign in the formula of undesirable outputs and is under the variable returns to scale (VRS) assumption because, under the constant returns to scale (CRS) assumption the abatement factor is equal to one. Based on the VRS assumption, however, there is a distinct difference between $\theta$ and $\theta^k$. The equation of the Kuosmanen technology is provided as (4). The difference between $P^K(x)$ and $P^S(x)$ is that the abatement factor changes from $\theta$ to $\theta^k$. Here, $P^K(x)$ satisfies the convexity axiom and has practicality in terms of abatement efforts. Figures 3 and 4 presents difference of estimated boundary and area caused by different abatement factors. Additionally, the difference between figures demonstrates the violation from Shephard technology. The shephard's method does not include an area in its PPF, in which it must be included along with an axiom4, convexity. Figures 3 and 4 present PPFs generated from DMU B and C based on Shephard's and Kuosmanen's method. The difference between them is whether they have $\overline{CH}$ or not. Shephard's PPF employs a single abatement factor($\theta$) and does not include $\overline{CH}$ and an area by it which comes from $\overline{CB}$ in Kuousmanen's by multiple abatement factors ($\theta^k$). $\Delta BDH$ is able to produce from B. Then, if H is possible to produce, $\overline{CH}$ and its area should be included in its PPF. However, in Figure 3, $\overline{CH}$ and its area are not included in its PPF. Therefore, Shephard's method violates an axiom4. This difference comes from abatement factors in each model. Multiple abatement factors in Kuousmanen's method cause inclusion of $\overline{CH}$ and its area, while a single abatement factor does not. More explanation is provided in Kuosmanen and Podinovski (2009) [18].

The Kuosmanen technology is a PPF that corresponds to the definition of weak disposability and satisfies all axioms. The Kuosmanen technology is the correct minimum extrapolation technology (The minimum extrapolation technology is the smallest production possibility set that satisfies all given axioms. Banker, Charnes, and Cooper (1984) generate this principle [28]. This is introduced from Kusomanen and Podinovski (2009) [18]), although the technology has a larger area than the Shephard technology because the Shephard technology has a violation and is limited. To address the violation and limitations, the Kuosmanen technology employs an equality and multiple abatement factors. Conventional models have limitations caused by PPFs, which are regarded as constraints in the respective models. To address these limitations, this paper proposes a model that employs an equality and multiple abatement factors.

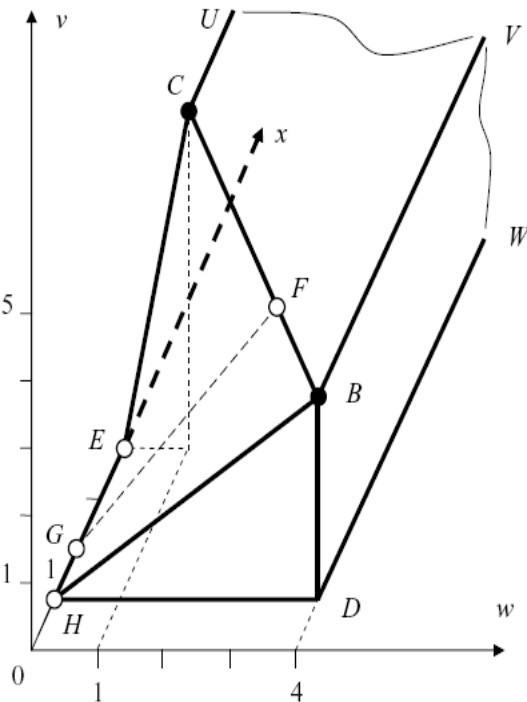

**Figure 3.** The Shephard technology induced by activities B and C. Source: Kuosmanen and Podinovski (2009).

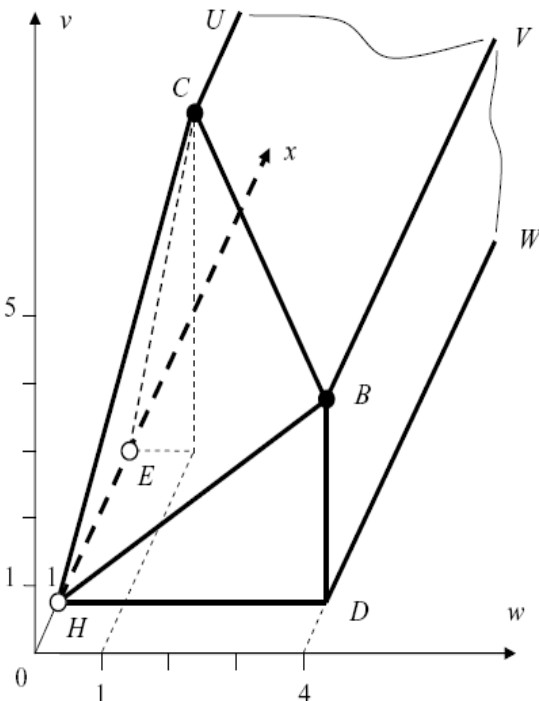

**Figure 4.** The Kuosmanen technology induced by activities B and C. Source: Kuosmanen and Podinovski (2009).

### 3.3. The Proposed Model

The majority of measurements using undesirable output model have employed DDF and SBM approach, particularly in aviation field. However, as mentioned in the previous section, the use of the DDF contradicts with the assumption of weak disposability. The proposed model takes the SBM approach because it is appropriate to reflect characteristics of undesirable outputs in terms of that it

does not have constrained directions to find the benchmark point. In addition, SBM approach allows to observe difference between subject and benchmark point by giving the specific amount of gap with slack. The PPF is defined based on Shephard (1970, 1974), and axioms are the same as those of Shephard (Section 3). From the perspective of the Kuosmanen technology, the proposed model employs multiple abatement factors and the VRS is assumed in the measurement (the reason is the same as that of Lozano and Gutiérrez (2011), which mentions that "given the limited competition among the airports, it cannot be expected that they operate at the most productive scale size." This is introduced in Banker (1984) [4,28]). The proposed model is shown as follows:

$$\text{Minimize } \rho = \frac{1-\left(\frac{1}{N}\right)\sum\limits_{n=1}^{N}\left(\frac{s_n^-}{x_n}\right)}{1+\left(\frac{1}{M}\right)\sum\limits_{m=1}^{M}\left(\frac{s_m^+}{v_m}\right)}$$

$$\begin{aligned}
\text{Subject} \quad &\text{to} \\
&\sum_{k=1}^{K} \theta^k z_k v_{km} - s_m^+ = v_m, m = 1,\ldots, M, \\
&\sum_{k=1}^{K} \theta^k z_k w_{kj} = w_j, j = 1,\ldots, J, \\
&\sum_{k=1}^{K} z_k x_{kn} + s_n^- = x_n, n = 1,\ldots, N, \\
&\sum_{k=1}^{K} z_k = 1, \ 0 \le \theta \le 1, \\
&z_k, s_m^+, s_n^- \ge 0, \forall k, m, n.
\end{aligned} \tag{5}$$

$k, m, j, n, v_{km}, w_{kj}, x_{kn}, v_m, w_j, x_n, \theta^k$: same explanation as Equation (4)
$s_m^+$: slack of desirable output
$s_n^-$: slack of input

Unlike conventional models using the SBM approach, Equation (5) has weak disposability between desirable and undesirable outputs by setting no slack in the equation of undesirable outputs and abatement factors. To correct the error from a single abatement factor, the abatement factor is represented by multiple abatement factors in Equation (5). Then, the model has a practical meaning that airports create different abatement efforts under the VRS assumption when the level of production activity is reduced. In this model, when a DMU find a benchmark point, it is based on its undesirable outputs. The difference between the conventional approach and the proposed models about undesirable outputs is directionality to find a benchmark point. In terms of outputs, the conventional one has independent directionality, which allows to increase desirable outputs and to decrease undesirable outputs, respectively. The proposed model finds frontiers, which have the same level of undesirable outputs with more desirable outputs. The latter has more practical perspective to find DMU's benchmark point than the former because it is almost impossible to increase production with less pollution without technological innovations, but in evaluation analysis, we usually assume that there are no such technological inventions.

## 4. Empirical Results (Applications to Airports' Eco-Efficiency)

### 4.1. Data and Analyses

The proposed model is used to measure the efficiency of 13 Korean airports (for consistency in dataset, Muan and Yangyang airports, which opened recently or temporarily, stopped operation are excluded.) for the 2004–2013 period. (Efficiency of Korean airports, along with other Northeast Asian airports, has also been studied in, e.g., Ha et al. (2010) and Ha et al. (2013) [29,30].). But these studies did not address the issue of undesirable outputs.) The details of airports are described in Figure 5

and Table 1. Seven variables are included in the evaluation. The length of the runway (in meters), the number of employees (in persons), and the terminal area (in square meters) are considered as input factors; the number of passengers (in persons), the amount of cargo (in tons), and the number of flights are considered as desirable outputs; and the level of $CO_2$ emissions (in tons of $CO_2$) is included as an undesirable output. To construct "smoothed-surface" frontiers in each analysis, a measurement requires 14 observations for the seven variables (Yu, 2004 [19]), but there are only 13 observations in each year. Therefore, the validity of efficiency cannot be guaranteed. To address this problem, the three-year-window DEA method (this method is recommended by Nghiem and Coelli (2002) [31].) is adopted. This method provides efficiency trends and stability over time as well as supplements the number of observations. In this analysis, the first window (or analysis) includes the 2004–2006 period, and the last window includes the 2011–2013 period. Therefore, a total of 39 observations are included in each analysis.

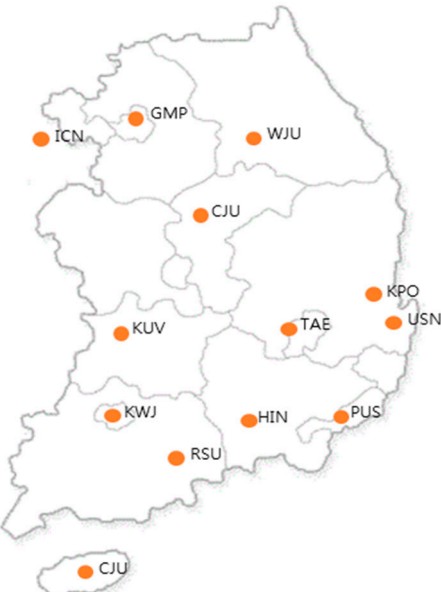

**Figure 5.** Map of Korean Airports.

**Table 1.** Korean Airports.

|  | Name (Airport) | Location (City) | Domestic/International |
|---|---|---|---|
| KWJ | Gwangju | Gwangju | Domestic |
| KUV | Gunsan | Gunsan | Domestic |
| GMP | Gimpo | Seoul | International |
| PUS | Gimhae | Busan | International |
| TAE | Daegu | Daegu | International |
| HIN | Sacheon | Sacheon | Domestic |
| RSU | Yeosu | Yeosu | Domestic |
| USN | Ulsan | Ulsan | Domestic |
| WJU | Wonju | Wonju | Domestic |
| ICN | Incheon | Seoul/ Incheon | International |
| CJU | Cheongju | Cheongju | International |
| CJJ | Jeju | Jeju | International |
| KPO | Pohang | Pohang | Domestic |

## 4.2. Comparisons between the Models

As we mentioned, there are two important issues which need to be considered in specific model. If all perspectives or technologies are compared at once by only using results, the comparison have pointless conclusion. In order to draw meaningful conclusion of the comparison induced from difference, it should be compared between perspectives, which have the same conditions. In that

way, the perspectives and comparisons are arranged in terms of each issue. According to two issues discussed previously, an empirical comparison is discussed.

### 4.2.1. A Comparison between the Lozano-Gutiérrez Model and the Shephard Model

Similar to the theoretical comparison in Section 3.1, we implement an empirical comparison about the weak disposability issue, using Korean airport's case. Additionally, through empirical comparison, the previous debate extends.

The model using the Lozano and Gutiérrez technology, that is, the Lozano and Gutiérrez model, considers undesirable outputs as inputs not as a variable for weak disposability. The difference between conditions of the equality and inequality in the equation of undesirable outputs can be shown by comparing the results for the Lozano and Gutiérrez model and the Shephard model:

The Lozano and Gutiérrez model is shown as Equation (7) (Unlike in the case of Equation (6), Lozano and Gutiérrez (2011) includes no x (input) in the objective function [4]. However, to compare technologies, this paper includes x in the objective function, as in Equation (6).). The model using the Shephard technology, that is, the Shephard model, is shown as Equation (6). Both models have a single abatement factor. The difference between these two models is the slack of w or inequality in the equation of undesirable outputs:

$$\text{Minimize } \rho = \frac{1-\left(\frac{1}{N}\right)\sum\limits_{n=1}^{N}\left(\frac{s_n^-}{x_n}\right)}{1+\left(\frac{1}{M}\right)\sum\limits_{m=1}^{M}\left(\frac{s_m^+}{v_m}\right)}$$

$$
\begin{aligned}
\text{Subject} \quad &\text{to} \\
&\theta \sum_{k=1}^{K} z_k v_{km} - s_m^+ = v_m, m = 1, \ldots, M, \\
&\theta \sum_{k=1}^{K} z_k w_{kj} = w_j, j = 1, \ldots, J, \\
&\sum_{k=1}^{K} z_k x_{kn} + s_n^- = x_n, n = 1, \ldots, N, \\
&\sum_{k=1}^{K} z_k = 1, \ 0 \le \theta \le 1, \\
&z_k, s_m^+, s_n^- \ge 0, \forall k, m, n.
\end{aligned}
\tag{6}
$$

$$\text{Minimize } \rho = \frac{1-\left(\frac{1}{N+J}\right)\left\{\sum\limits_{n=1}^{N}\left(\frac{s_n^-}{x_n}\right)+\sum\limits_{j=1}^{J}\left(\frac{s_j^-}{w_j}\right)\right\}}{1+\left(\frac{1}{M}\right)\sum\limits_{m=1}^{M}\left(\frac{s_m^+}{v_m}\right)}$$

$$
\begin{aligned}
\text{Subject} \quad &\text{to} \\
&\theta \sum_{k=1}^{K} z_k v_{km} - s_m^+ = v_m, m = 1, \ldots, M, \\
&\theta \sum_{k=1}^{K} z_k w_{kj} + s_j^- = w_j, j = 1, \ldots, J, \\
&\sum_{k=1}^{K} z_k x_{kn} + s_n^- = x_n, n = 1, \ldots, N, \\
&\sum_{k=1}^{K} z_k = 1, \ 0 \le \theta \le 1, \\
&z_k, s_m^+, s_n^-, s_j^- \ge 0, \forall k, m, n, j.
\end{aligned}
\tag{7}
$$

$k, m, j, n, v_{km}, w_{kj}, x_{kn}, v_m, w_j, x_n$: same explanation as Equation (4)

$s_m^+, s_n^-$: same explanation as Equation (5)

$s_j^-$: slack of undesirable output

These appear in the production possibility area. There is a proportional line created by the abatement factor θ which lies in the front areas of the Shephard technology and Lozano and Gutiérrez technology graphs. The front areas have the same shape because they share something in common—the same abatement factor. The rear parts have different areas because of differences between equations of w. Spatially, the Lozano and Gutiérrez model in Equation (7) has a larger area because of the inequality, which implies that production can infinitely produce undesirable outputs like inputs. The argument of Haliu and Veeman that weak disposability and equality greatly inflate the efficiency score is considered in the same vein. However, the gap in efficiency between the Lozano and Gutiérrez model and the Shephard model is generally insignificant. Figures 6 and 7 show the results for the measurement of efficiency based on the models. Figure 6 shows similar trends in efficiency, which implies that the two models show minor differences in efficiency scores over time. Figure 7 shows a chart of average efficiency by airport. Airports show slight differences between the models but there is no pattern for the differences. For example, Yeosu Airport (RSU) and Incheon International Airport (ICN) have higher efficiency scores under the Shephard model than under the Lozano and Gutiérrez model, but Gimpo International Airport (GMP) and Sacheon Airport (HIN) show the opposite results. Some airports such as Ulsan Airport (USN) show the same efficiency score between the models.

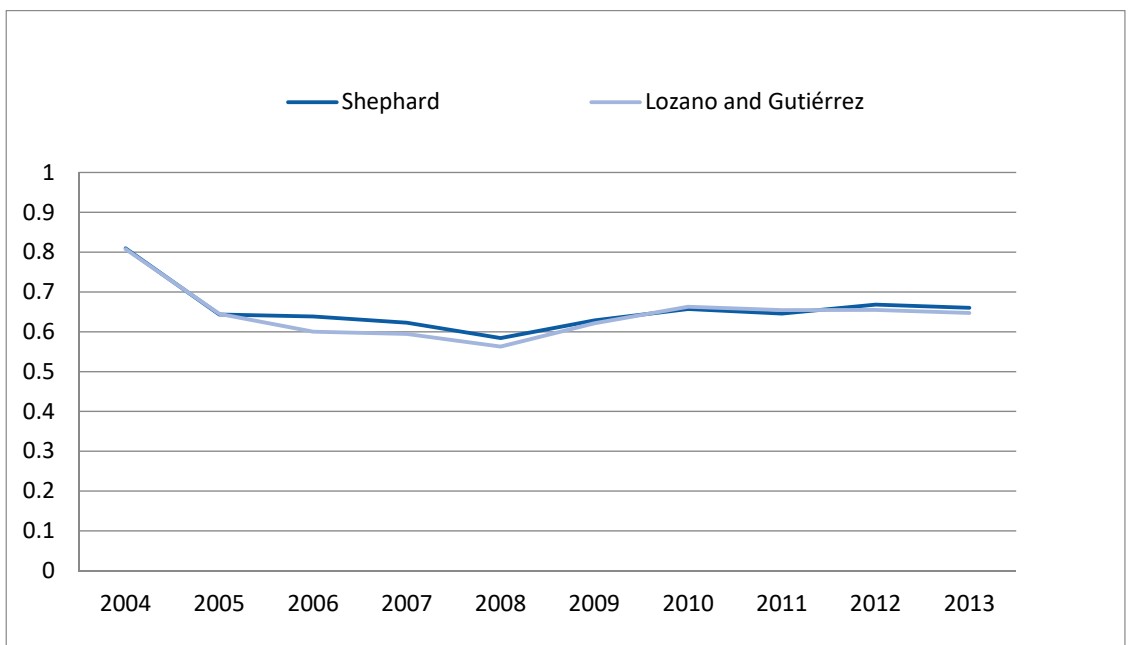

**Figure 6.** A comparison of trends in model efficiency by year.

The Lozano and Gutiérrez model has the slack of undesirable outputs ($CO_2$ emissions), and therefore the slack influences the efficiency score. This is consistent with the perspective of Haliu and Veeman, who assert that weak disposability, which has an equality in the equation of w, produces an undetermined effect of undesirable outputs on efficiency. If this is the case, then there must be a significant difference between the technologies. However, a comparison of the results shows no significant difference between the Lozano and Gutiérrez model and the Shephard model. This confirms that undesirable outputs have considerable influence on efficiency by influencing the projecting direction of an observation to any benchmark point for a reference set in weak disposability, although it is not included in the objective function.

Table 2 provides the difference between models, by comparing the number of being benchmark in two models. Especially ICN presents significant different between models. This difference is caused by different PPFs and direction to find benchmark point. The perspective of an undesirable output as an input is erroneous and limited theoretically and practically. Based on this perspective, Lozano and

Gutiérrez attempt to reflect weak disposability in the model. The Lozano and Gutiérrez model has the same limitations because of the slack of undesirable outputs, which can decrease independently. In addition, the limitations of weak disposability introduced by Haliu and Veeman are not valid with respect to empirical results. Therefore, it is desirable to take weak disposability into account in an undesirable-output model by this equality.

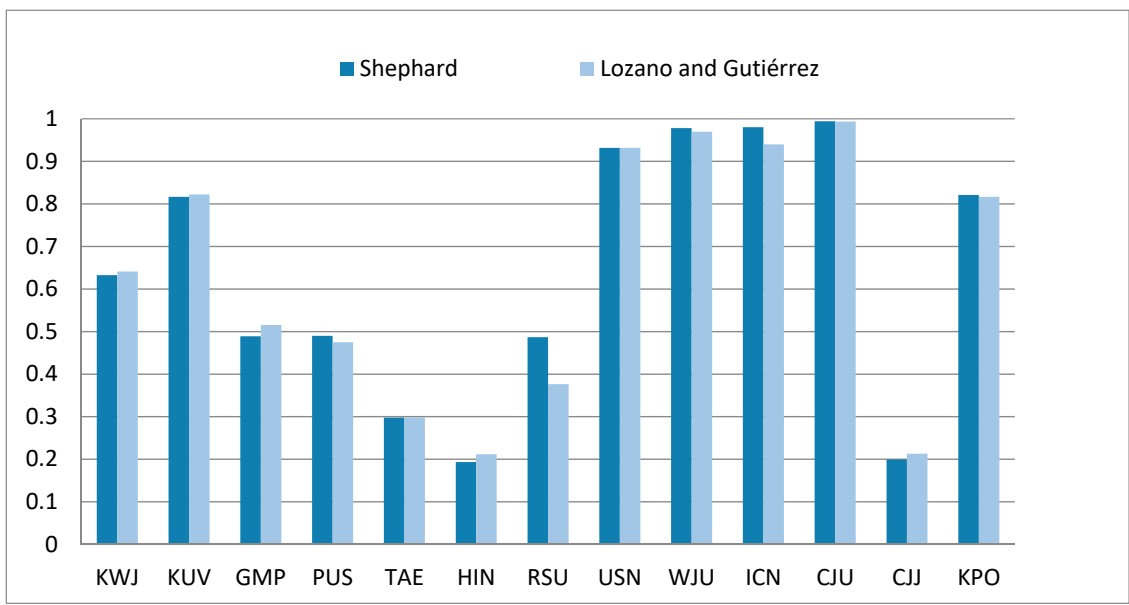

**Figure 7.** A comparison of model efficiency by airport.

**Table 2.** The number of being referenced by DMUs.

|  | KWJ | KUV | GMP | PUS | TAE | HIN | RSU | USN | WJU | ICN | CJU | CJJ | KPO |
|---|---|---|---|---|---|---|---|---|---|---|---|---|---|
| Shephard | 9 | 16 | 2 | 0 | 0 | 1 | 29 | 52 | 172 | 137 | 199 | 0 | 43 |
| Lozano & Gutiérrez | 6 | 21 | 2 | 0 | 0 | 1 | 15 | 54 | 176 | 82 | 210 | 0 | 47 |

### 4.2.2. A Comparison of the Shephard Model and the Proposed Model

Based on the discussion on the abatement factor issue in Section 3.2, Shephard model and the proposed model are compared empirically.

Figure 8 compares the proposed model to the Shephard model by considering weak disposability in the model and showing the efficiency trends. The proposed model has lower scores than the Shephard model for consecutive years. This verifies that multiple abatement factors include the front part of the area of the PPF, which a single abatement factor does not include, and that this found area influences the efficiency score. Figure 9 compares efficiency scores by airport. In general, the proposed model has lower efficiency scores. In particular, Gunsan Airport (KUV) shows a large difference in the efficiency score between the models. Table 3 shows the difference between the models. If the difference is less than 0, then the proposed model has a lower efficiency score than the Shephard model. Gunsan Airport (KUV) and Sacheon Airport (HIN) change by approximately 105% and 36%, respectively. These changes are caused by multiple abatement factors. This verifies that multiple abatement factors allow a technology to include an area that is not included in a technology using a single abatement factor.

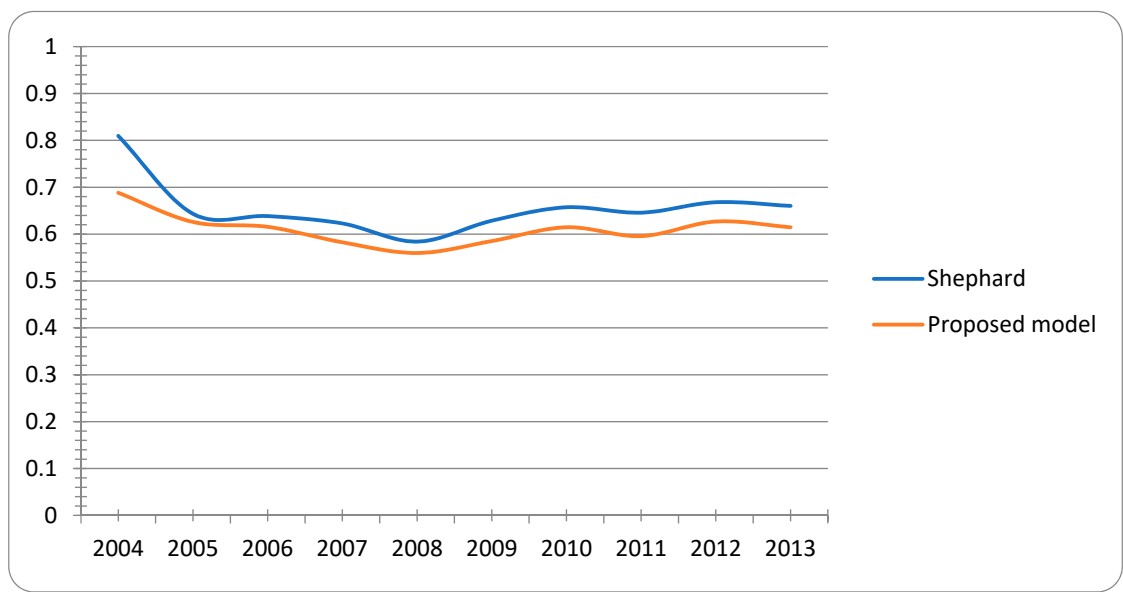

**Figure 8.** A comparison of efficiency trends.

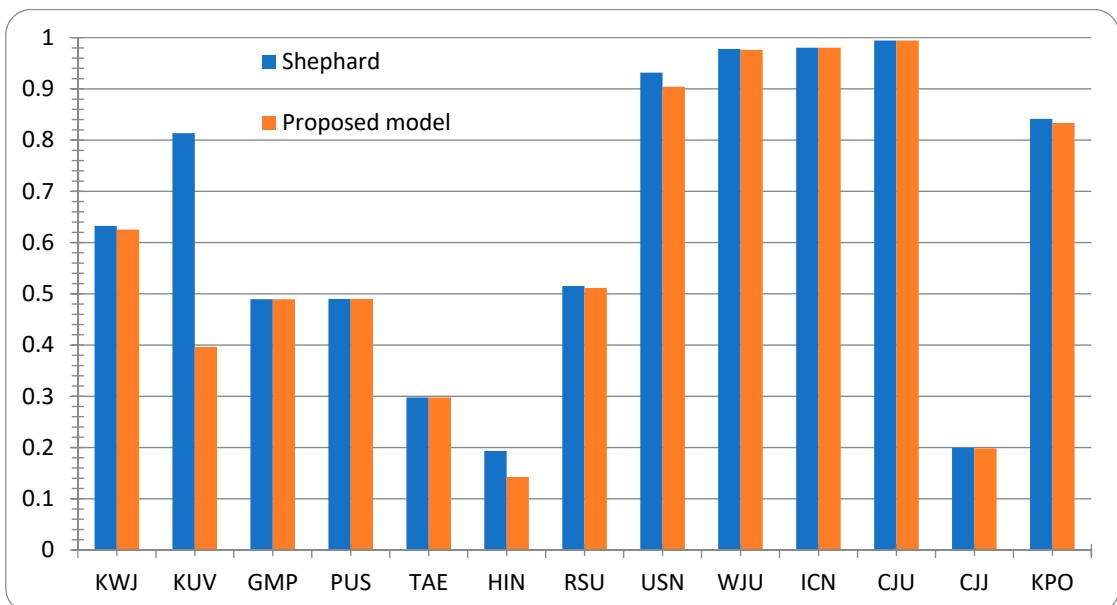

**Figure 9.** A comparison of efficiency scores by airport.

**Table 3.** Gaps in efficiency scores between models.

|  | KWJ | KUV | GMP | PUS | TAE | HIN | RSU | USN | WJU | ICN | CJU | CJJ | KPO |
|---|---|---|---|---|---|---|---|---|---|---|---|---|---|
| Gap | −0.0070 | −0.4171 | 0.0000 | 0.0000 | 0.0000 | −0.0508 | −0.0036 | −0.0275 | −0.0019 | 0.0000 | 0.0000 | −0.0011 | −0.0080 |
| Change ratio (%) | −1.12 | −105.28 | 0.00 | 0.00 | 0.00 | −35.68 | −0.71 | −3.04 | −0.20 | 0.00 | 0.00 | −0.58 | −0.95 |

Notes: Gap: average efficiency of the proposed model − average efficiency of Shephard model (by airports), Change rate: gap/average efficiency of the proposed model (by airports).

The SBM approach provides the amount of slack, which indicates the ability to improve production, and therefore a comparison of the model's slack is meaningful. Table 4 shows the average change in this slack. This change is calculated by dividing the difference in the slack score between the models (gap) by the mean value of observed data of each airport. This difference is the absolute value of the difference in average slack between the Shephard model and the proposed model. The change in cargo slack is about 1.4 times larger for Gunsan Airport (KUV) than for the observed data, and for Sacheon

Airport (HIN), it is about three times the observed data. Ulsan Airport (USN), Gunsan Airport (KUV), and Sacheon Airport (HIN) also show large changes.

**Table 4.** Average change ratios in slack between models (%).

|  | Runway Length (Meter) | Terminal Size (Meter sq.) | # of Employees (Persons) | Passenger (Persons) | Cargo (Tons) | Air Movement (Flights) |
|---|---|---|---|---|---|---|
| KWJ | 0.3 | 3.8 | 2.6 | 5.7 | 7.5 | 4.0 |
| KUV | 37.0 | 2.5 | 16.3 | 43.5 | 138.1 | 32.7 |
| GMP | 0.0 | 0.0 | 0.0 | 0.0 | 0.0 | 0.0 |
| PUS | 0.0 | 0.0 | 0.0 | 0.0 | 0.0 | 0.0 |
| TAE | 0.0 | 0.0 | 0.0 | 0.1 | 0.0 | 0.0 |
| HIN | 5.6 | 16.9 | 3.4 | 59.2 | 301.7 | 25.4 |
| RSU | 0.0 | 0.0 | 0.0 | 1.5 | 9.6 | 0.7 |
| USN | 0.3 | 0.0 | 0.1 | 2.8 | 83.5 | 0.0 |
| WJU | 0.2 | 0.0 | 0.0 | 0.2 | 0.1 | 0.5 |
| ICN | 0.0 | 0.0 | 0.0 | 0.0 | 0.0 | 0.0 |
| CJU | 0.0 | 0.0 | 0.0 | 0.0 | 0.0 | 0.0 |
| CJJ | 0.4 | 3.0 | 2.1 | 9.9 | 14.8 | 7.2 |
| KPO | 0.0 | 0.0 | 0.0 | 0.2 | 6.9 | 0.1 |

Notes: Change rate: $\frac{\text{gap}(|\text{average slack of the proposed model} - \text{average slack of Shephard model}|)}{\text{mean value of observed data of an airport}}$ (by airports).

This is caused by a change in the benchmark point because the frontier is changed by multiple abatement factors. Based on a calculation of multiple slack values between the models, the cargo slack is approximately 3.6 times larger in the proposed model; the passenger slack is approximately 4.2 times larger; the air movement slack is approximately 4.8 times larger for Gunsan Airport (KUV), and the passenger slack is approximately 2.1 times larger for the Ulsan Airport (USN).

## 5. Conclusions

This paper has considered undesirable outputs to evaluate efficiency by comparing previous undesirable-output models, namely the Lozano and Gutiérrez model and the Shepard model, in the context of the aviation industry and proposes a new model. The Lozano and Gutiérrez model is based on the Haliu and Veeman model, which regards undesirable outputs as inputs and considers weak disposability in the model. Ha, H.K. (2011b) takes the SBM approach to present a similar perspective. This method allows undesirable outputs to decrease independently based on the slack of undesirable outputs, although the model has an abatement factor in the PPF. A single abatement factor makes variables contract in the same proportion but cannot make them move in the same direction to increase or decrease together. Therefore, the perspective of the Lozano and Gutiérrez model has a limitation in reflecting weak disposability. The Shephard technology assumes weak disposability as a property of undesirable outputs and provides a PPF using an equality in the equation of undesirable outputs ($w$) and a single abatement factor ($\theta$). Yu (2004), Pathomsiri et al. (2008), and Fan et al. (2014) follow this perspective. However, a single abatement factor causes practical and technical limitations. First, there is a practical problem with a single abatement factor implying that all airports make the same abatement effort if there is a reduction in production activity. This is not practical because each airport has its own manner of production such that there are differences in abatement efforts across airports. A single abatement factor causes a technical problem in that the convexity axiom is violated. The Shephard technology neglects the area or subset that should be included in the production possibility set by convexity [18]. To address these limitations and reflect weak disposability correctly, this paper proposes a model using an equality in the equation of undesirable outputs ($w$) and multiple abatement factors based on the SBM approach. The results provide no support for Haliu and Veeman's criticism and suggests that the equality reflects weak disposability based on a comparison of results between the Lozano and Gutiérrez model and the Shephard model. The results based on a comparison of the

Shephard model and the proposed model verify that multiple abatement factors in the latter have valid effects to overcome the limitations of a single abatement factor in conventional models.

In the long-term, weak disposability can be realized in another manner. Technically, the equality method has been the best way to consider weak disposability in many models until now. However, this method is not the ideal way to impose weak disposability on undesirable outputs because a fixed output with characteristics of by-products and harmful products can have practical limitations. In addition, there may be an increase in the efficiency scores without a general increment. In this regard, constructing better alternatives in considering weakly disposable undesirable outputs than the equality method will be needed. In future research, we also should analyze many cases to raise the significance of results and to secure diversity.

**Author Contributions:** Conceptualization, T.-W.Y. and H.-K.H.; Methodology, T.-W.Y. and H.-K.H.; Validation, T.-W.Y. and H.-K.H.; Formal Analysis, T.-W.Y. and J.-H.N.; Investigation, T.-W.Y.; Resources, T.-W.Y.; Data Curation, T.-W.Y.; Writing-Original Draft Preparation, T.-W.Y.; Writing-Review & Editing, J.-H.N. and H.-K.H.; Visualization, T.-W.Y. and J.-H.N.; Supervision, H.-K.H.; Project Administration, H.-K.H.

**Funding:** This research was supported by Inha University.

**Conflicts of Interest:** The authors declare no conflicts of interest.

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
