# Peer review of "Comparative Analysis of Production Possibility Frontier in Measuring Social Efficiency with Data Envelopment Analysis: An Application to Airports"

_sustainability, doi:10.3390/su11072181_

Round 1
Reviewer 1 Report
This paper provides both theoretical and empirical contribution to the literature regarding the efficiency measurement where the production process involves an undesirable output. It comprehensively discusses the methodology of the productivity assessment with bads, while adopting the method to the actual data of Korean airports. I see the paper has meaningful contribution to the literature, thus justifies its publication in Sustainability.
Author Response
Thank you very much for your comments.
Reviewer 2 Report
The paper evaluate airport efficiency with respect to environmental factors by comparing Lozano-Gutierrez, Shepard, and the proposed undesirable-output model with slack-based measure approach with empirical data from Korea’s 13 domestic airports. The authors suggests that proposed model validly overcome the limitations of single abatement factors in the other models.
The manuscript contains some grammar issues which make the paper really hard to read and very annoying, suggest to do a thorough editorial review by a professional spell and grammar editor. Few examples, Line 34 , Delete “All”. Line 34-35, “greenhouse gases (GHGs) emission” should be “greenhouse gas (GHG) emissions”. Line 35, “are the issue which is the most actively discussed.” Should be “are most actively discussed issues” Line 37 “ are the main objective” should be “ is the main objective” and ect. Line 123 and line 279 delete “Here”
Line 61, explain what the axiom that the model violated is.
Line 268 to 269, Can undesirable outputs influence the efficiency score under the proposed method? Please clarify
Equation (5) should be below line 269 statement.
Line 278, can the author clarify the reason that the data selected is from 2004 to 2013 considering the publication will be in 2019? If there is new data available can the authors use the newly available data say from 2014 to 2018 to verify the results?
Line 278, and 284, can the author find another airport data in Korean? Or can the airport data from other country included in the analysis. 13 sounds like a small data set. Please clarify and explain.
Line 353, under this section the authors conclude the change between the two method by comparing their efficiency scores. I would like to see a statistical significant test result to support the conclusion that the two model performs differently.
Author Response
Thank you very much for your kind and detailed comments.
The manuscript contains some grammar issues which make the paper really hard to read and very annoying, suggest to do a thorough editorial review by a professional spell and grammar editor. Few examples, Line 34 , Delete “All”. Line 34-35, “greenhouse gases (GHGs) emission” should be “greenhouse gas (GHG) emissions”. Line 35, “are the issue which is the most actively discussed.” Should be “are most actively discussed issues” Line 37 “ are the main objective” should be “ is the main objective” and ect. Line 123 and line 279 delete “Here”
- answer : we corrected all the articles by taking all your comments.
Line 61, explain what the axiom that the model violated is.
- answer : We explained it in line 249-259, and also briefly stated in line 236-238.
Line 268 to 269, Can undesirable outputs influence the efficiency score under the proposed method? Please clarify Equation (5) should be below line 269 statement.
- answer : We explained it in line 286-294.
Line 278, can the author clarify the reason that the data selected is from 2004 to 2013 considering the publication will be in 2019? If there is new data available can the authors use the newly available data say from 2014 to 2018 to verify the results?
- answer : This research cannot include more years in dataset due to lack of the nember of employee data. Korean airports employ a lot of non-regular workers. For recent years, an inequality issue between regular and non-regular workers has become a serious social problem, and so airport operators avoid to provide their employee information.
Line 278, and 284, can the author find another airport data in Korean? Or can the airport data from other country included in the analysis. 13 sounds like a small data set. Please clarify and explain.
- answer : For consistency in dataset, we excluded some airports which recently opened or temporarily suspended such as Muan and Yang-Yang. To cover pointed weakness, we employed the three-year-window DEA method as stated in line 305–309.
Line 353, under this section the authors conclude the change between the two method by comparing their efficiency scores. I would like to see a statistical significant test result to support the conclusion that the two model performs differently.
- answer : We could not apply statistical significant test due to lack of data on diverse cases and leaved it for the future research. Neverthless, the significance of the result in this analysis cannot be denied because the proposed model is based on proved theoretical concepts.
Reviewer 3 Report
Dear author,
Your paper is rather confusing. And for the reader that are not in your field of study your paper is very hard to understand. Also, I could no see the advantages of your proposed model in relation to the other models, that must be more clearer to the reader.
Regards,
Reviewer.
Author Response
Thank you very much for your kind comments.
comments
Your paper is rather confusing. And for the reader that are not in your field of study your paper is very hard to understand. Also, I could no see the advantages of your proposed model in relation to the other models, that must be more clearer to the reader.
- answer : To evaluate airports’ efficiency considering the environmental factor, conventional researches on the efficiency analysis attempted to reflect characteristics of weak disposability, but they have limitations in terms of PPFs and directionality to find benchmark points. We analyzed the references associated with this filed with two perspectives and found out which one is the reasonable way to calculate the environmental efficiency score with Korean airports’ case. We expected that the research results of this research contribute for exact assessment of airports’ productivity in the environmental view as well as understanding the present state of comprehensive performance of airport industry.
We have modified the unclear parts of the whole paper to reflect your comments.
Reviewer 4 Report
This paper shows some interesting research results. The author(s) emphasized their research mainly on airport efficiency analysis, undesirable outputs and on evaluating productivity with respect to environmental factors. From the viewpoint of research contributions, it holds the real findings of the proposed research topic. However, the contents and descriptions of the empirical results (applications to airports’ eco-efficiency) in this paper are a little vague that the reader is hardly to understand how the research is operated appropriately. A necessary expansion and reworking of that part in this paper should restructure this article for a well publication ready format. The authors are encouraged to revise the paper (as per the items discussed below) and then afterward to reconsider for possible publication and to make an interesting contribution to the journal.
The author(s) need to explain why the two models (i.e., the Shephard technology vs. Lozano-Gutiérrez model) could or need been compared with each other?
The results indicate that the proposed model with the slack-based measure (SBM) approach evaluates eco-efficiency better than conventional models. But how the SBM is related to the above mentioned two models? What are their relationships or their differences existing between them? Please explain the details. Cause devil is truly in the details.
The discussion of the empirical example, i.e., the Korean airports, stated in this paper is not clear enough. Some more detailed descriptions may be needed to show exactly what and where (via more Maps or Figures etc.) the complete characteristics of the empirical example should be.
For the purpose of research validation and verification use, the authors should try to verify their proposed method by showing the objective outcomes of this project, say, via the official evidences from government or others.
In my view, the paper has the potential to make an interesting contribution to the journal. However, while the underlying research is sound, the detailed descriptions of the research work and writing style for an academic journal still needs some work to be done which could make the article complete.
Author Response
Thank you very much for your useful comments.
comment 1: The author(s) need to explain why the two models (i.e., the Shephard technology vs. Lozano-Gutiérrez model) could or need been compared with each other?
Answer: We explained it in line 56-65.
comment 2: The results indicate that the proposed model with the slack-based measure (SBM) approach evaluates eco-efficiency better than conventional models. But how the SBM is related to the above mentioned two models? What are their relationships or their differences existing between them? Please explain the details. Cause devil is truly in the details.
Answer: We explained it in line 70-72.
comment 3: The discussion of the empirical example, i.e., the Korean airports, stated in this paper is not clear enough. Some more detailed descriptions may be needed to show exactly what and where (via more Maps or Figures etc.) the complete characteristics of the empirical example should be.
Answer: We reflected this comment in figure 5 and table 1.
comment 4: For the purpose of research validation and verification use, the authors should try to verify their proposed method by showing the objective outcomes of this project, say, via the official evidences from government or others.
Answer: There are no rigid indicators taking CO2 into account when they evaluate airports’ performance, especially in Korea. We try to give a standard in assessing productivity with environmental outputs through this research.
Round 2
Reviewer 2 Report
The revisions are satisfied
Author Response
We revised the paper
Reviewer 3 Report
Dear Author,
You did an extensive work, however your results did not present a sound difference for the traditional models.
Author Response
Unfortunately we cannot apply our model to other diverse cases due to lack of data, and leave it for the future research. Nevertheless, we think that the significance of the result in this analysis cannot be denied because the proposed model is based on proved theoretical concepts.